# Relapse-Free Survival and PD-L1 Expression in First High- and Low-Grade Relapsed Luminal, Basal and Double-Negative P53-Mutant Non-Muscular Invasive Bladder Cancer Depending on Previous Chemo- and Immunotherapy

**DOI:** 10.3390/cancers12051316

**Published:** 2020-05-21

**Authors:** Ekaterina Blinova, Dmitry Enikeev, Dmitry Roshchin, Elena Samyshina, Olga Deryabina, Aleksander Tertychnyy, Dmitry Blinov, Evgenia Kogan, Marina Dudina, Haydar Barakat, Dmitrij Merinov, Aleksandr Kachmazov, Stanislav Serebrianyi, Natalia Potoldykova, Dmitrij Perepechin

**Affiliations:** 1Department of Clinical Anatomy and Operative Surgery, Department of Pathological Anatomy, Institute for Urology and Reproductive Health, Sechenov University, 8/1 Trubetzkaya Street, Moscow 119991, Russia; bev-sechenov@mail.ru (E.B.); dvenikeev@gmail.com (D.E.); atertychnyy@yandex.ru (A.T.); koganevg@gmail.com (E.K.); a5pfvsis@yandex.ru (M.D.); potoldykovanv@gmail.com (N.P.); 2Department of Oncological Urology, Russian National Research Center of Radiology, 3 2nd Botkinsky Proezd, Moscow 125284, Russia; dr89031990702@gmail.com (D.R.); d.merinov@gmail.com (D.M.); kac.68@mail.ru (A.K.); volecon@mail.ru (S.S.); medcraft@mail.ru (D.P.); 3Laboratory of Molecular Pharmacology and Drug Design, All-Union Research Center for Biological Active Compounds Safety, 23 Kirova Street, Staraja Kupavna 142450, Russia; samy-elena@yandex.ru; 4Department of Oncology, Laboratory of Pharmacology, National Research Ogarev Mordovia State University, 68 Bolshevistskaya Street, Saransk 430005, Russia; dep-general@adm.mrsu.ru; 5Department of Bioinformatics, People’s Friendship University of Russia, 6 Miklukho-Maklaya Street, Moscow 117198, Russia; dr.haydarbarakat@yahoo.com

**Keywords:** non-muscular invasive bladder cancer, tumor grade, molecular subtype, relapse, PD-L1 expression, survival, malignant cells, immune cells

## Abstract

The goal of this study was to assess how PD-L1 expression in tissue specimens of patients with main molecular subtypes of NMIBC (luminal, basal and double-negative p53-mutant) associates with relapsed-free survival in dependence on the tumor grade and prior treatment of primary bladder cancer. PD-L1 expressions on the membrane of neoplastic and CD8+ immune cells were assessed in tumor specimens (*n* = 240) of primary and relapsed luminal, basal and double-negative p53-mutant NMIBC. Association between relapse-free survival and PD-L1 expression was estimated for high- and low-grade relapsed NMIBC according to previous treatment and their molecular profile, using the Kaplan–Meier method, and assessed by using the log-rank test. Potential confounders were adjusted by Cox regression models. In a group of patients who underwent only TUR without intravesical therapy, there were significant differences in relapse time between high- and low-grade tumors in basal and luminal molecular subtypes; for basal relapsed carcinoma, RFS was shorter in cases where tumors were less malignant. Both intravesical mitomycin and Bacillus Calmette–Guerin (BCG) therapy significantly extended the time of recurrence of low-grade luminal and basal bladder malignancies with no intergroup differences in double-negative NMIBC. PD-L1 expression status was associated with RFS for luminal relapsed NMIBCs in the group without previous frontline intervention, and with RFS in the group of patients with luminal relapsed bladder cancer previously utilized BCG. Obtained results may be considered as a promising approach for further clinical implementation.

## 1. Introduction

Despite the extensive information and many previous studies on this topic, in some patients with non-muscular invasive bladder cancer (NMIBC), the disease progression occurs in an unpredictable manner. It is obvious that a natural history of disease in a late follow-up period (more than 5 to 10 years) cannot be assessed only by clinical or patho-morphological features of the primary tumor [1]. At the present time, a simple EORTC’s stratification based on tumor type or its grade failed to answer the question of the possible chemotherapy and/or immunotherapy efficacy, as well as to predict the progression of the disease in a late follow-up period [2]. Nowadays, programmed death receptor ligand 1 (PD-L1) is the most studied immune checkpoint molecule, which promotes immunosuppression by binding to PD-1 on T-cells in tumor immunity, and it already plays a crucial role in muscle-invasive bladder cancers (MIBC) treatment [3,4]. In NMIBC patients, its value remains unclear [5]. It is, therefore, necessary to investigate the possible potential relationship between PD-L1 expression and Bacillus Calmette–Guerin (BCG) immunotherapy effectiveness, as well as the prognostic significance of PD-L1 expression in association with other molecular-genetic markers, such as basal and luminal NMIBC subtypes. The two intrinsic subtypes of bladder cancer show different clinical behavior and responses to frontline chemotherapy [2,6]. In chemotherapy naive setting, NMIBC of the basal subtype was more aggressive, with shorter survival, when compared to luminal non-muscle invasive bladder cancer. On the other hand, basal bladder cancer was more sensitive to particular chemotherapy, and the patients with this form of the disease have more benefits from frontline chemotherapy than luminal subtype [6]. The prognostic value of basal or luminal tumor types in association with PD-L1 expression in NMIBC patients has not been evaluated yet. At the same time, some investigators reported that bladder cancer expressing high PD-L1 showed a poor prognosis [7,8,9], but others suggested high PD-L1 level predicted the good prognosis [10]. Thus, according to Kawahara et al. (2018), correlation between PD-L1 expression and the prognosis remains controversial [11]. The aim of this study was to assess how PD-L1 expression in tissue specimens of patients with main molecular subtypes of NMIBC (luminal, basal and double-negative p53-mutant) associates with relapsed-free survival in dependence on the tumor grade and prior treatment of primary bladder cancer.

## 2. Results

Four groups of patients were created, representing individual tumorigenic history as follows: primarily diagnosed urothelial carcinoma, first relapsed chemotherapy- and immunotherapy-naive non-muscular bladder cancer, and two groups of first relapsed tumors after either mitomycin or BCG intravesical treatment. Each group includes tumors of main molecular subtypes, according to Dadhania et al. [2] signature, as follows: GATA3(+)/KRT5/6(−) (luminal molecular subtype), GATA3 (−)/KRT5/6(+) (basal molecular subtype) and GATA3(−)/KRT5/6(−) (double-negative p53- expressing molecular subtype) (Appendix A, Figure A1). In turn, every molecular subgroup was divided into high- and low-grade malignancies based on histological examination [12] and World Health Organization (WHO) 2004/2016 classification [13].

### 2.1. PD-L1 Expression in Primary and Relapsed Non-Invasive Urothelial Tumors

Primary carcinomas expressed PD-L1 differently depending on the tumor molecular subtype and its grade (Figure 1). Primary luminal low-grade tumors’ PD-L1 expression level was 19.6 ± 1.3%, while in more malignant tumors, it was 29.3 ± 1.3% (*p =* 0.001). Expression level of firstly detected basal NMIBC of high-grade was 5.0 ± 0.6% and increased up to 8.4 ± 0.6% in the case of low-grade tumors. Double-negative p53-mutant primary high-grade urothelial cancer was characterized by scantily PD-L1 expression (2.0 ± 0.4%), whereas tumors with low malignant potential had zero-Median PD-L1-expressing profile.

In relapsed tumors of patients left without any kind of intravesical chemo- and immunotherapy after transurethral resection of primary urothelial carcinoma, we observed unevenness of PD-L1 expression. Luminal and basal highly malignant non-invasive carcinomas showed intensive membrane anti-PD-L1-staining occupied 26.7 ± 0.7% and 24.7 ± 0.6% of cells, respectively. Low-grade NMIBC specimens of luminal molecular subtype were less positively stained than more malignant ones (14.8 ± 1.1%, *p* = 0.005). Double-negative p-53 expressing relapsed tumors of both grades contained an equally low number of PD-L1-expressing cells (high-grade tumor—1.0 ± 0.5%; low-grade tumor—0.8 ± 0.2%).

Recurrent tumors after previous transurethral resection and following intravesical instillation of mitomycin were generally characterized by low PD-L-1 expression. In high- and low-malignant GATA3-expressing tumors, 9.8 ± 0.5% and 7.8 ± 0.7% of cells expressed PD-L1 (*p =* 0.06). We observed differences in the molecular marker expression level between potentially more-malignant KRT5/6-expressing NMIBC (5.2 ± 0.7%) and a less-malignant one (0.8 ± 0.5%, *p* = 0.001). Registration of positively stained cells in double-negative p53-mutant bladder cancer sections also revealed intergroup significance between high-grade (4.3 ± 0.7%) and low-grade (2.0 ± 0.4%, *p* = 0.005) tumors.

According to our observation, frontline immunotherapy with BCG caused sufficient impact on PD-L1-expressing status of afterward relapses of bladder tumor. Thus, both advanced and low-malignant luminal recurrent NMIBC showed a high marker expression rate (36.0 ± 2.0% and 24.1 ± 1.2%, respectively), though with intergroup differences (*p* = 0.005). For basal molecular subtype of bladder cancer recurrence, PD-L1 expression profile of both high- and low-grade specimens was significantly low—21.0 ± 1.7% and 6.9 ± 1.0%, respectively (*p* = 0.001, intergroup comparison), whereas double-negative p53-mutant highly malignant relapsed tumors exhibited 40.8 ± 3.2% of positively-stained cells (*p* = 0.001, in comparison with the low-malignant subgroup, where only 16.8 ± 1.1% of tumor cells expressed PD-L1).

### 2.2. Intensity of Tumor-Associated Immune Cells’ Infiltration in Primary and Relapsed NMIBC

As it has been reported, tumor microenvironment is broadly involved in cancer development and progression [14,15,16,17,18]. In relation to the subject of our study, tumor-infiltrating immune cells in urothelial cancers can express PD-L1 exhibiting immune downregulation, which should be taken into consideration when assessing what kind of cells—malignant or stromal—expresses PD-L1. Therefore, we assessed the level of T-suppressor population in each of the study groups (Figure 2).

In primary bladder carcinoma CD8+ cells occupied 26.7 ± 1.4% of tumor area of high-grade luminal NMIBC and 16.8 ± 0.8% (*p =* 0.001) of tumor tissue with low malignant potential; basal molecular subtype of primary bladder cancers presented a low level of CD8+ expression, which was significantly higher in low-grade tumors (8.0 ± 0.8%) than in highly malignant tissues (5.3 ± 0.7%; *p =* 0.005). Double-negative p53-expressing primary tumors of both grades were equally infiltrated by T-suppressor immune cells. They occupied no more than 2.8 ± 0.4% of high-grade and 2.4 ± 0.4% of low-grade tumor area. High-grade relapsed chemotherapy- and immunotherapy-naive tumors of both luminal and basal molecular subtypes were intensely infiltrated by immune CD8+ cells: T-suppressors occupied 24.9 ± 1.3% and 24.1 ± 1.0% of the tumor area, respectively, while CD8+ expression in low-grade tumors was 13.4 ± 1.2% for luminal subtype (*p =* 0.002 in comparison with high-grade GATA3-positive NMIBC) and 19.5 ± 1.3% for basal one. Double-negative p53-mutant relapsed bladder carcinoma reproduced the same level of immune cells’ tumor-related population as it was in luminal and basal NMIBCs, without intergroup differences between high- and low-grade subtypes.

The first relapse of NMIBC after intravesical mitomycin treatment was generally characterized by a low level of CD8+ expression and demonstrated no differences between carcinomas with high and low malignant potential. On the contrary, recurrent tumors of patients treated with intravesical BCG broadly differed regarding both the neoplasm molecular subtype and malignant potential. Thus, 26.8 ± 1.8% of cells of high-grade GATA3-positive relapsed bladder cancer expressed CD8+; in low-grade tumors, 21.7 ± 1.5% of cells represented stromal immune microenvironment. KRT5/6-positive highly malignant urothelial carcinoma was infiltrated by immune cells at 17.1 ± 1.4% level, whereas in low-grade basal NMIBC, CD8+ cells occupied only 5.1 ± 0.8% of tumor area (*p =* 0.001). Unlike all groups described above, 29.7 ± 1.2% of tumor area of high-grade double-negative p-53 mutant subtype of recurrent NMIBC was occupied by CD8+-expressing cells; more than twice lower T-suppressor population (12.3 ± 0.9%; *p =* 0.001 intergroup comparison) infiltrated relapsed double-negative p53-expressing bladder carcinoma with low malignant potential.

### 2.3. Mapping of PD-L1 Expressing Status of Relapsed NMIBCs

As we have mentioned above, to assess tumor PD-L1 expression status correctly, it is crucial to determine whether neoplastic or stromal immune cells are positively stained with anty-PD-L1 antibody. We estimated PD-L1 expression level for malignant and CD8+ immune cells in each study subgroup as per the antibody manufacturer’s interpretation guide [19], and we presented the results as a color-map (Figure 3).

All primary low-grade and luminal high-grade tumors had Low PD-L1-expressing status, while in malignant basal and double-negative cancers, High PD-L1+ status was related to prevalent immune cells’ membrane expression. Tumor-associated microenvironment determined High PD-L1 status of GATA3-positive and KRT5/6-expressing bladder tumors detected as the first relapse after TUR without frontline intravesical instillations. On the contrary, double-negative p53-mutant tumors of this group did not intensely react with anti-PD-L1 antibody. In the mitomycin-utilized group of relapsed tumors, low-malignant basal and double-negative recurrent bladder carcinomas had Low PD-L1+ status. Immune cells of luminal and more malignant types of basal and double-negative p53 mutant tumors were highly PD-L1-expressive. Among the first recurrent tumors that emerged after previous BCG intravesical immunotherapy, high-grade luminal and double-negative bladder carcinomas reproduced High PD-L1+ status. Notably, that expression level was determined in both stromal and neoplastic cells.

### 2.4. Association Between PD-L1 Expression and Relapse-Free Survival in Chemotherapy/Immunotherapy-Naive and Frontline Treated NMIBC of Main Molecular Subtypes with High and Low Malignant Potential

For each molecular subtype of recurrent NMIBC, we evaluated the associations between the tumor grades and relapsed-free survival (Figure 4). In the group of patients who had not utilized frontline treatment after TUR, the Kaplan–Meier plot showed that relapse time was lower for those with high-grade luminal NMIBC. For the basal molecular subtype of bladder cancer, relapse time was lower for patients with low-grade urothelial carcinoma. In the group of patients with a double-negative p53-expressing molecular subtype of NMIBC, relapse time was lower for those with high-grade tumors. According to the long-rank test, there was significant evidence of a difference in relapse times in luminal and basal NMIBC for low- and high-grade tumors (*p* < 0.05).

In the group of patients that utilized intravesical mitomycin after TUR, the emerging time of high-grade luminal relapsed tumor was lower than the time to low-malignant urothelial cancer recurrence. In the case of high-grade basal NMIBC, relapse-free survival was lower in comparison with low-grade tumors. Survival time to double-negative p53 mutant NMIBC recurrence was not associated with the tumor grade. According to long-rank test, there was significant evidence of a difference in relapse times in luminal and basal NMIBC for low- and high-grade relapsed carcinomas (*p* < 0.05).

TUR followed by intravesical BCG instillations led to increase relapse-free time to recurrence of low-grade luminal and basal NMIBC with intergroup RFS significance between low- and high-grade relapsed tumors (*p* < 0.05, long-rank test). For double-negative p53-expressing NMIBC, relapse time was the same for patients with high- and low-grade bladder malignancies.

With the assistance of univariate analysis, we found out that PD-L1 expression status was associated with relapse time of luminal NMIBC in the group of bladder cancer without previous frontline intervention, and it was also associated with RFS in the group of patients with luminal relapsed bladder cancer previously utilized frontline therapy by BCG instillations (Table 1). At the same time, PD-L1 expression was not associated with RFS in the group of patients with first relapsed basal NMIBC that previously underwent only TUR (*p* = 0.053) and in the group of BCG-treated NMIBC with relapsed double-negative p53-expressing urothelial cancer (*p* = 0.5). We did not find a link between PD-L1 expression status and RFS for patients with high- and low-grade NMIBC relapses of all molecular subtypes whose primary tumors were treated by intravesical mitomycin.

## 3. Discussion

Different studies have been initiated to explore difficult and ambiguous relationship among a broad range of biological, social, gender and other factors involved in development, progression and treatment response of non-muscular invasive bladder cancer [1,2,4,5,6,7,8,9,10,11]. Tumor behavior is supposed to be a result of the neoplasia’s nature, its molecular subtype and malignant potential, which in sophisticated combinations increase predictive uncertainty and leave practitioners without strong prognosticators. The discovery of immune avoidance as a major mechanism of cancer growth and progression revolutionized therapeutic approaches in oncology [20]. Programmed cell death receptor and its ligand 1 form a molecular axis, along which NMIBC, amid other tumors, breaks natural immune defense [7,8,11]. Therefore, inhibition of PD-1/PD-L1 signaling might be a promising way to control the disease’s aggressive behavior. At the same time, molecular multiplicity of the tumor influences the surgical and therapeutical outcomes of anticancer interventions [9,10]. Taking into consideration molecular and malignant diversity of primary and relapsed bladder cancer, the idea of our study was to analyze how PD-L1 expression associates with relapse-free survival of patients with recurrent GATA3 (+), KRT5/6 (+) and double-negative NMIBC regarding tumor grade and previous surgical and chemo-/immunotherapy settings.

We found out that first basal and luminal relapses of NMIBC display upregulation of PD-L1-expression (except low-grade basal mitomycin-treated NMIBC) regardless of the tumor grade and previous therapy. On the contrary, high-grade double-negative p53-mutant recurrent NMIBC expressed PD-L1 differently in dependence on previously utilized treatment, whilst low-grade double negative bladder relapsed tumors were always PD-L1 low or negative. Davick with co-authors previously reported upregulation of PD-L1 expression in high-grade urothelial carcinoma, but without association with overall survival in study groups [20]. The authors highlighted particularities of PD-L1 regulation largely in primary bladder tumors. Earlier we documented the same PD-L1 expressing pattern in GATA3 (+), KRT (+) and p53-mutant human-derived xenograft tumors of urothelial origin [21]. It is utterly important to determine a nature of PD-L1 positivity for each molecular and malignant kind of relapsed urothelial carcinoma, because targeting of PD-1/PD-L1 pathway on the membrane of neoplastic cells and on CD8+ T-suppressor cells possesses powerful potency for effective disease control [22]. We observed that tumors of all groups were infiltrated by PD-L1-expressing CD8+ cells with prevalence of T-suppressor population (more than 10% of all cells) in chemotherapy- and immunotherapy-naive high- and low-grade relapsed luminal and basal NMIBC, and in high- and low-grade luminal and double-negative, high-grade basal relapsed urothelial carcinoma after previously utilized frontline immunotherapy by BCG. A similar level of immune infiltrates in T1 high-grade tumors versus MIBC was reported by Wankowitz et al. [23]. The researchers underlined that prevalence of immune cell PD-L1 positivity was characterized for non-invasive T1 urothelial cancer, whereas in muscular invasive bladder tumors, the authors detected a high number of positively stained neoplastic cells. Similar observations were documented for ovarian and breast cancers, as well as melanoma [18,24,25]. In our study, we determined that PD-L1 expression in recurrent non-invasive bladder tumors was associated with not only tumor grade, but also with NMIBC molecular subtype and previously utilized frontline treatment.

It is widely known that frontline treatment of primary NMIBC influences clinical outcomes and relapse-free survival, in particular [9]. Nevertheless, an association between previous treatment, molecular subtype of first relapsed NMIBC and its PD-L1 expression status remains unclear. We found out that, in the group of patients that underwent only TUR without intravesical frontline therapy, there were significant differences in the time to relapse between high- and low-grade tumors in basal and luminal molecular subtypes. Interestingly, for basal relapsed carcinoma, RFS was shorter in the case of less-malignant tumors. Survival to relapses with high and low malignant potential did not differ in double-negative bladder cancer. Only for luminal (GATA3-positive) relapses in this group time to relapse was associated with PD-L1 expression status. Intravesical mitomycin therapy significantly prolonged the time to recurrence of low-grade luminal and basal bladder malignancies with no intergroup differences in double-negative NMIBC. There was not any link between PD-L1 expression status and RFS for relapsed tumors in this group of patients. TUR with subsequent intravesical instillation of BCG postponed an occurrence of low-grade luminal and basal relapsed NMIBC, with strong association between PD-L1 expression status and RFS of patients with luminal molecular subtype of NMIBC relapse.

## 4. Materials and Methods

### 4.1. Study Design

The study design is summarized in Figure A2 (Appendix B). The morphological diagnosis and tumor grades of all malignancies were evaluated by histological examination (HE) of hematoxylin and eosin-stained sections, according to the WHO classification guidelines. After that, tumors were classified into CIS, Ta, T1, T2, T3 and T4 [12,13]. We took chemo-/immunotherapeutic background from medical records for each case of relapsed bladder cancer. Initially, for the purpose of differentiating basal, luminal and double-negative p53-mutant subtypes of primary NMIBC and relapsed tumor GATA3, KRT5/6 and p53 expression were determined by immunohistochemistry in formalin-fixed and paraffin-embedded samples. Regarding the obtained diagnostic data, all cases were divided in twelve groups, as follows: luminal primary NMIBC, basal primary NMIBC, double-negative p53-mutant primary NMIBC; first relapsed luminal, basal and p53-mutant double-negative chemotherapy- and immunotherapy-naive NMIBC, first relapse of luminal, basal and p53-mutant double-negative tumors previously treated with intravesical mitomycin or BCG, with 20 cases in each group. Taking into consideration a role of tumor microenvironment in cancer progression and reply to healing intervention, level of CD8+ immune cells was evaluated for each tumor sample. After that, specimens of all groups underwent immunohistochemical testing for PD-L1 expression. Finally, association between relapse-free survival and PD-L1 expression was estimated for high- and low-grade relapsed NMIBCs, according to previous treatment and molecular profile.

### 4.2. Ethic Statement

Research Protocol of this non-interventional study and patient’s consent form were reviewed and approved independently by Ethic Committee of Sechenov University at the Committee Board meeting on September 16, 2019 (Approval No. 9; Rev. No 16/09-1-2019) and Scientific Board of Bio-Ethic Commission of National Research Medical Center of Radiology (Approval No. 12, 02 October 2019; Rev. No. 02/10-2019-E).

### 4.3. Data Sources and Study Population

#### 4.3.1. Sample Size and Study Population

To date, the data of PD-L1 expression level in NMIBC are limited and controversial. There are a few studies in which researchers evaluated PD-L1 expression in different NMIBC variants, using different diagnostic approach, clones of antibody and expression cutoff [20,26,27,28]. We calculated two variants of sample size estimation based on minimal and maximal quantity data, using the Chi-squared statistic (or z-test). Due to the absence of exact data about PD-L1 expression in basal and luminal NMIBCs, we calculated the sample size based on data from general urothelial cancer population. In the articles used as the basis for the calculation, some authors assessed PD-L1 expression in tumor cells and the cutoff for PD-L1 positivity was ≥ 5%. In one previous study, PD-L1 expression on tumor cells after BCG treatment became positive at 70%. We were planning to assess the PD-L1 expression in both tumor and immune cells and used the 25% of PD-L1 positive cells as cutoff. Considering the information listed above, we assumed that the sample size would be near to a mean value between these two calculations. We used G*Power 3.1.9.4 for Mac OS X 10.7 (HHU, Dusseldorf, Germany) to estimate sample size for the study.

An informed consent for using tissue samples and disease-related information was voluntarily obtained from each study participant. In total, 240 patients diagnosed with non-muscular invasive bladder cancer who underwent cystoscopy with biopsy and/or transurethral resection with appropriate 6-year follow-up period at Sechenov University and Russian National Research Medical Center of Radiology between 9 January 2014 and 30 December 2019 were enrolled in the study. The index date for participants was their diagnosis date. Table 2 displays most important disease-related characteristics of patients enrolled in the study. All patients met the eligibility criteria listed below.

The inclusion criteria consisted of the following:Age > 18 years at the time of first diagnosis;Diagnosis of NMIBC;With available formalin-fixed tumor tissue sample for IHC testing.

Detection of muscular-invasive bladder cancer as primary tumor and a patient’s death before NMIBC relapse were exclusive criteria for the study.

Study population consisted of 140 males with average age of 59.6 ± 3.8 and 100 females aged 61.3 ± 2.9. The majority of all primary and relapsed cases displayed T1 urothelial papillary carcinoma with low-grade (total number—176) and high-grade (total number—61) malignant potential, according to WHO 2004/2016 classification [13]. Three tumors, one of which from the group of primary-detected neoplasms and the other two from relapsed BCG-treated group, were diagnosed as high-grade monofocal primary CIS histologically presented by micropapillary carcinoma and squamous cancer. Overall gender and grade-related intergroup patients’ distribution was homogenous among primary, relapsed untreated and relapsed treated groups (Table 2).

#### 4.3.2. Data Sources

Primary formalin-fixed and paraffin-embedded tumor specimens (*n* = 240) were obtained from Tissue Banks of National Research Medical Center of Radiology and Institute of Urology of Sechenov University, between 2014 and 2019. Tissue conditions met basic requirements for biological samples’ preserving; this ensured equal diagnostic quality of all samples.

### 4.4. Immunohistochemistry

For ICH, we used 4 µm thick histological sections of surgical or biopsy material of urothelial bladder cancer. All tissue samples were fixed in a buffered 10% formalin solution for 6–72 h, dehydrated and saturated with paraffin, according to the standard protocol in tissue histological Spin vacuum processor (STP250-V, “Histo-Line Laboratories Srl”, Italy) in automated regimen. All paraffin-embedded blocks contained enough material to study protein expression by using both positive and negative controls.

We used monoclonal antibody against human GATA3 (HG3-31 clone, dilution, 1:100; Santa Cruz Biotechnology Inc., Santa Cruz, CA, USA), to evaluate transcription factor encoded by GATA3 gene expression and classify bladder cancer as belonging to luminal molecular subtype. To select basal molecular subtype of non-invasive urothelial carcinoma, we stained the sections with monoclonal antibody against human KRT5/6 (D5/16B4 clone, 1:50 dilution, Dako, Denmark). We used >80% cutoff for KRT5/6 cytoplasm staining with low/undetectable (<10%) GATA3 nuclear positivity to classify urothelial cancer into basal molecular subtype, and vice versa for luminal subtype of NMIBC [29,30]. To assess CD8+ immune cell population in tumor specimens, we counted positively stained cells, using anti-CD8 antibody (catalog No. ab4055, Abcam, Cambridge, UK).

To determine the tumor samples’ PD-L1-expressing status we operated with Ventana PD-L1 (SP263) Assay with the OptiView DAB IHC Detection Kit (Cat. No. 760–700/06396500001) and signal amplification (Ventana Medical Systems, Inc., Tucson, AZ, USA). From each tumor sample, two sections were made for research and for negative control. In the negative control, the section was stained according to the manufacturer’s protocol, similarly to the main study; however, the VENTANA Rabbit Monoclonal Negative Control Ig (Cat. No. 790–4795/06683380001) negative control reagent was used as a primary antibody. We used tonsils tissue samples as a positive control in each reaction cycle. To set up a reaction with the SP263 assay, after the dewaxing and unmasking antigens, rabbit monoclonal antibody PD-L1 (SP263) in working dilution was applied to the prepared tumor sections. After incubation with the primary antibody or negative control reagent, the OptiView DAB IHC Detection Kit with OptiView Amplification Kit was used. The final step was hematoxylin staining. All reaction steps were carried out in an automated mode on a VENTANA BenchMark ULTRA ICH stainer (Ventana Medical Systems, Inc., Tucson, AZ, USA), as per manufacturer’s instructions. At the first stage of the analysis, the staining of positive and negative cell lines and positive tissue control was evaluated. Then, ICH reactions were evaluated for urothelial cancer samples. The percentage of PD-L1-positive immune and tumor cells among the total number of cells was scored in five randomly PC-selected high-power fields (magnification, ×500), and percentages ≥ 25% of tumor cells (TC) and/or ≥ 25% of immune cells (if they (ICP) represented more than 1% of total cell population) with positively stained membranes (IC+) were classified as High PD-L1+ status. The positive membrane staining rate was between 0% and 25% for TC or/and between 0% and 25% for IC+, if ICP ≥ 1% of all cell population was considered as Low PD-L1+ status. Absence of membrane positive staining appropriated negative PD-L1 status.

### 4.5. Statistical Analysis

Descriptive statistics (Mean ± SD), median and frequencies summarized tumor laboratory characteristics for all enrolled subjects. The following variables were analyzed: primary variable—relapse-free survival, and laboratory variables PD-L1 expression status, CD8+ expression status, grade of PD-L1 expression. For statistical analysis SPSS (version 22.0, IBM, Chicago, IL, USA) was used, and *p* < 0.05 was considered statistically significant. To study the differences of PD-L1 expression (high or low expressing status) among the study groups, independent *t*-test was used. The associations between PD-L1 expression and relapsed-free survival were estimated using the Kaplan–Meier method and assessed by using the log-rank test. Potential confounders were adjusted by Cox regression models, with the PD-L1 fitted as indicator variables.

## 5. Conclusions

Ultimately, our obtained results illuminate differences in relapse-free survival of patients with luminal, basal and double-negative NMIBC relapses of high- and low-malignant grade, regarding previously utilized treatment. The association between PD-L1 expression status and time to relapsed luminal recurrent urothelial carcinoma of patients who had not received frontline therapy and those treated with BCG may be considered as a promising approach to further clinical implementation. The role of other biological markers, such as miRs and FGFR3 gene expression, in relapsing of different molecular subtypes of NMIBC would be an interesting subject for our future study.

## Figures and Tables

**Figure 1 cancers-12-01316-f001:**
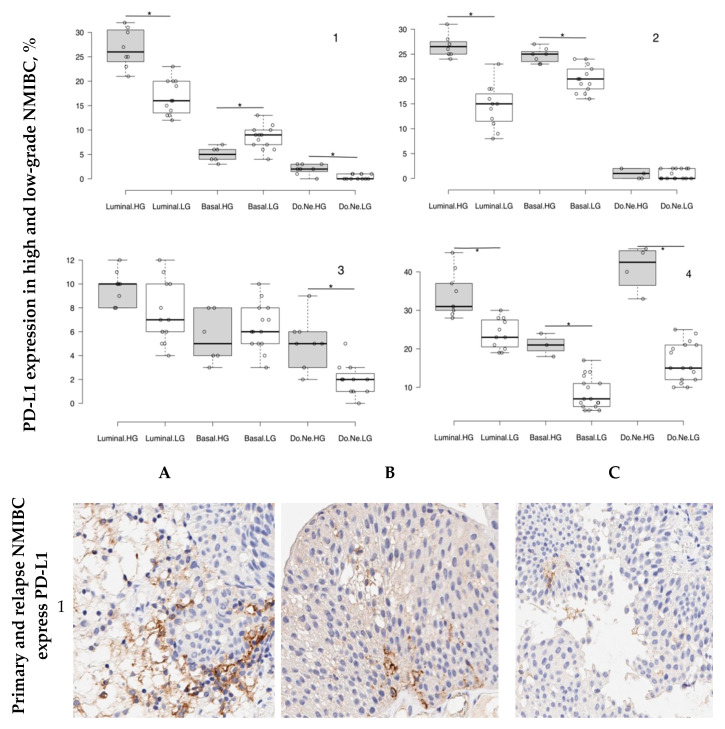
PD-L1 expression in primary and recurrent NMIBC. Boxplots show Medians of percentage of anti-PD-L1 positively stained cells in high-grade (HG) and low-grade (LG) primary (1), relapsed untreated (2), relapsed mitomycin-treated (3) and Bacillus Calmette–Guerin (BCG)-utilized (4) tumor specimens of luminal molecular subtype of bladder cancer (**A**); basal subtype of urothelial carcinoma (**B**) and double-negative (Do.Ne.) p53-expressing NMIBC (**C**): ***—*p* < 0.05, intergroup comparison between HG and LG tumors of the same molecular subtype, independent *t*-test. Microphotographs of anti-PD-L1-stained tumor sections, *IHC*, ×500.

**Figure 2 cancers-12-01316-f002:**
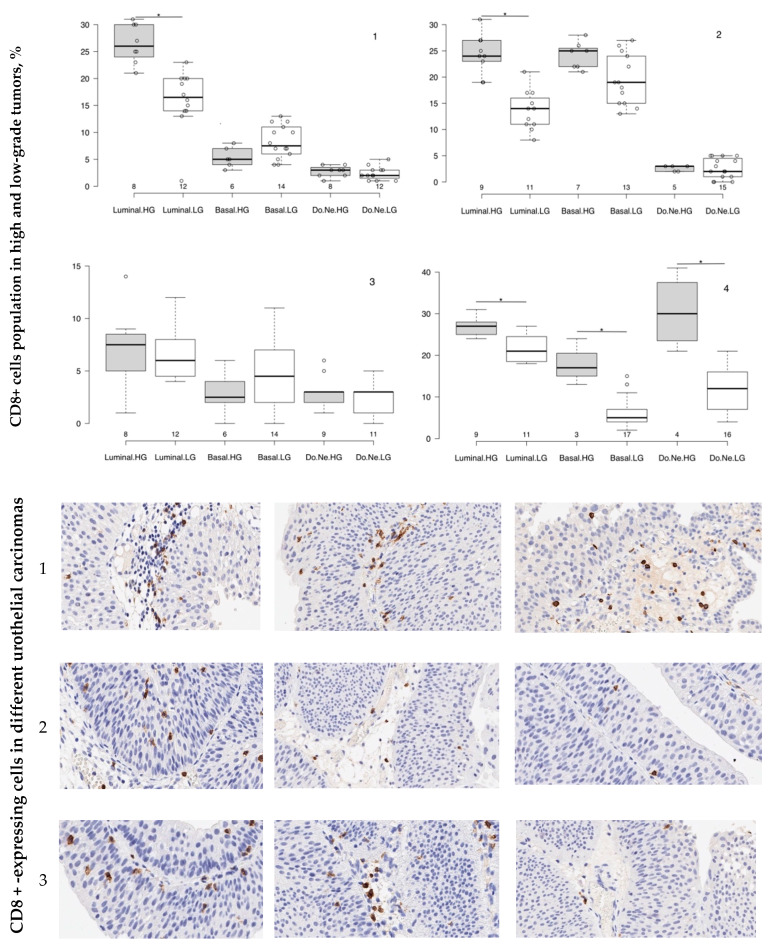
Tumor-associated immune CD8+ cells infiltration in primary and relapsed NMIBCs. Boxplots summarized primary data and Medians of CD8+ expression in high-grade (HG) and low-grade (LG) luminal (**A**), basal (**B**) and double-negative (Do.Ne.) p53-expressing (**C**) primary (1), relapsed untreated tumors (2) and recurrent urothelial cancers of patients utilized intravesical mitomycin (3) and BCG (4): ***—*p* < 0.05, intergroup comparison between HG and LG tumors of the same molecular subtype, independent *t*-test. Microphotographs display brown-colored nuclei of anti-CD8+ *IHC-*stained immune cells, ×500.

**Figure 3 cancers-12-01316-f003:**
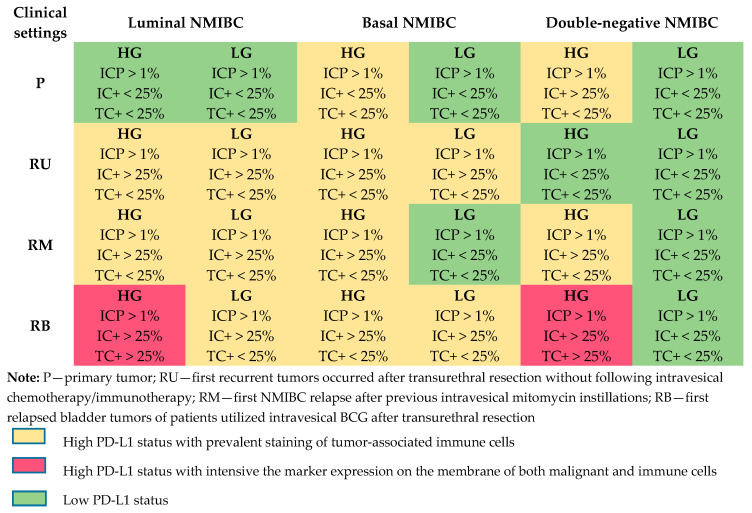
High and Low PD-L1 status of primary and relapsed NMIBC.

**Figure 4 cancers-12-01316-f004:**
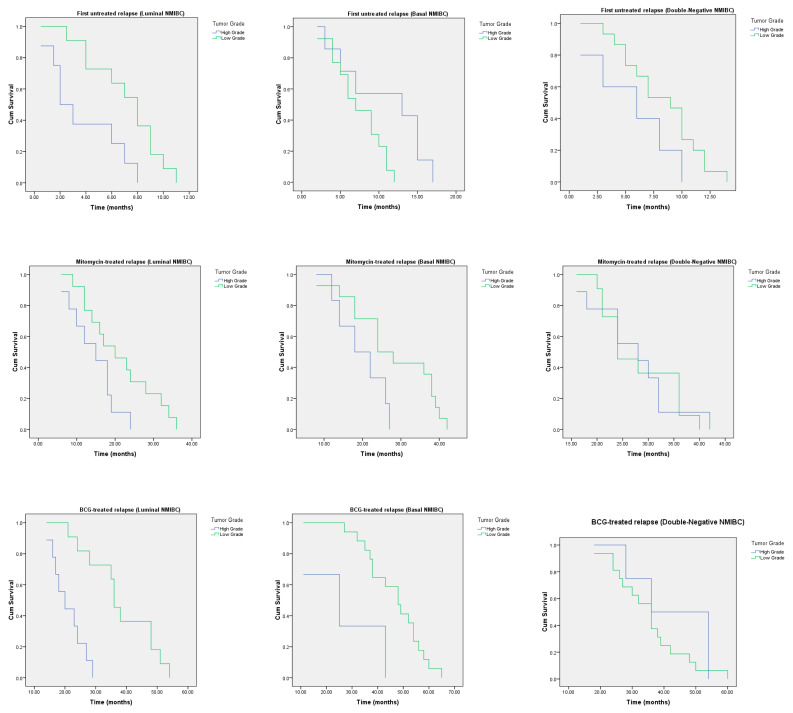
Kaplan–Meier actuarial analysis of relapse-free survival time in groups of patients with High- and Low-grade of luminal, basal and double-negative p53-expressing molecular subtypes of NMIBC. Plots depict cumulative survival (Cum Survival) separately for each grade and molecular subtype of the tumor, in association with previously utilized frontline treatment.

**Table 1 cancers-12-01316-t001:** Univariate Cox regression analysis of association between PD-L1 expression status (High vs. Low) and relapse-free survival.

Relapse	Subgroups	HR	95% CI	*p*-Value
**First relapse, chemotherapy- and immunotherapy-naive settings**	Luminal NMIBC	0.29	0.10–0.83	0.022
	Basal NMIBC	3.59	0.98–13.14	0.053
	Double-negative NMIBC	-	-	-
**First relapse, mitomycin-treated settings**	Luminal NMIBC	-	-	-
	Basal NMIBC	-	-	-
	Double-negative NMIBC	-	-	-
**First relapse, BCG-treated settings**	Luminal NMIBC	0.04	0.006–0.37	0.004
	Basal NMIBC	-	-	-
	Double-negative NMIBC	1.44	0.49–4.17	0.50

**Table 2 cancers-12-01316-t002:** Disease-specific characteristics of patients, according to their distribution to the study groups.

Study Group	Tumor Stage, Grade, n	Gender (n)	Age, Mean ± SD	Tumor Histology
Primary tumors, *n* = 20 in each group				
Luminal NMIBC	T1, HG, *n* = 8	M (3), F (5)	54.2 ± 4.1	Urothelial papillary carcinoma, micropapillary carcinoma
	T1, LG, *n* = 12	M (8), F (4)	57.6 ± 3.8	
Basal NMIBC	T1, HG, *n* = 6	M (3), F (3)	61.4 ± 5.3	
	T1, LG, *n* = 14	M (9), F (5)	60.2 ± 4.4	
Double-negative	CIS, HG, *n* = 1	M (0), F (1)	72	
	T1, HG, *n* = 7	M (4), F (3)	58.3 ± 2.9	
	T1, LG, *n* = 12	M (7), F (5)	55.9 ± 3.7	
Total	LG (41), HG (19)	M (34), F (26)	Average 57.7 ± 3.6	
Immunotherapy/chemotherapy-naive relapsed tumors, *n* = 20 in each group				
Luminal NMIBC	T1, HG, *n* = 9	M (6), F (3)	63.7 ± 4.1	Urothelial papillary carcinoma
	T1, LG, *n* = 11	M (8), F (3)	58.7 ± 4.6	
Basal NMIBC	T1, HG, *n* = 7	M (4), F (3)	64.3 ± 3.3	
	T1, LG, *n* = 13	M (8), F (5)	51.5 ± 4.5	
Double-negative	T1, HG, *n* = 5	M (3), F (2)	53.1 ± 5.4	
	T1, LG, *n* = 15	M (7), F (8)	54.6 ± 2.8	
Total	LG (39), HG (21)	M (36), F (24)	Average 55.4 ± 3.8	
First relapse after prior intravesical Mitomycin treatment, *n* = 20 in each group				
Luminal NMIBC	T1, HG, *n* = 8	M (4), F (4)	50.2 ± 3.2	Urothelial papillary carcinoma
	T1, LG, *n* = 12	M (7), F (5)	54.2 ± 2.7	
Basal NMIBC	T1, HG, *n* = 6	M (2), F (4)	64.0 ± 3.6	
	T1, LG, *n* = 14	M (11), F (3)	59.4 ± 3.5	
Double-negative	T1, HG, *n* = 9	M (3), F (6)	55.6 ± 4.8	
	T1, LG, *n* = 11	M (5), F (6)	54.7 ± 3.5	
Total	LG (37), HG (23)	M (32), F (28)	Average 56.4 ± 3.5	
First relapse after prior intravesical BCG treatment, *n* = 20 in each group				
Luminal NMIBC	CIS, HG, *n* = 2	M (2), F (0)	56; 67	Urothelial papillary carcinoma, micropapillary carcinoma, squamous cancer
	T1, HG, *n* = 7	M (4), F (3)	61.1 ± 3.2	
	T1, LG, *n* = 11	M (7), F (4)	58.9 ± 2.6	
Basal NMIBC	T1, HG, *n* = 3	M (2), F (1)	48.4 ± 4.5	
	T1, LG, *n* = 17	M (9), F (8)	53.2 ± 2.9	
Double-negative	T1, HG, *n* = 4	M (3), F (1)	55.9 ± 3.8	
	T1, LG, *n* = 16	M (9), F (7)	57.5 ± 3.0	
Total	LG (46), HG (14)	M (36), F (24)	Average 53.8 ± 3.4	

M—male; F—female; LG—low grade; HG—high grade.

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
