# Peer review of "Relapse-Free Survival and PD-L1 Expression in First High- and Low-Grade Relapsed Luminal, Basal and Double-Negative P53-Mutant Non-Muscular Invasive Bladder Cancer Depending on Previous Chemo- and Immunotherapy"

_cancers, 2020, doi:10.3390/cancers12051316_

Round 1

Reviewer 1 Report

The authors studied the relapse-free survival and PD-L1 expression in high grade and low grade relapsed luminal, basal and double-negative P53-mutant non-muscular invasive bladder cancer. The authors included sufficient number of cases and considered relevant clinicopathologic classifications. They found PD-L1 expression was associated with prognosis for certain types. 

I have only few comments.

The cut-off values of immunopositivity for CK5/6 and GATA3 to classify the tumors into basal and luminal types should be given.

The double negative tumors seem to be all p53-mutant as read from the menuscript. Is it true? Not all double negative tumors were p53-mutant in Dadhania's article.

The journal name and doi address of Dadhania's paper (ref. 2) are incorrect. It should be EBioMedicine,  http://dx.doi.org/10.1016/j.ebiom.2016.08.036 (not Oncotarget, ....033)

Author Response

Dear Reviewer,

On behalf of all the manuscript authors I thank you for valuable remarks, which we have used as effective guide to improve our article.

Point 1.

The cut-off values of immunopositivity for CK5/6 and GATA3 to classify the tumors into basal and luminal types should be given.

Reply 1.

According to Wang et al. [ref. 29] and Lerner et al. [ref. 30] we used > 80% cut-off for KRT5/6 cytoplasm staining with low/undetectable (< 10%) GATA3 nuclear positivity to classify urothelial cancer into basal molecular subtype, and vice versa for luminal subtype of NMIBC (lines 493-496).

Point 2.

The double negative tumors seem to be all p53-mutant as read from the manuscript. Is it true? Not all double negative tumors were p53-mutant in Dadhania's article.

Reply 2.

Your judgment is absolutely correct. Not all double-negative bladder cancers are p53-mutant.  But in our study we selected only p53-expressing GATA3 and CR5/6 negative tumors in prospect of our further (and now ongoing) study of prognostic, pathological role of some miRs expression and FGFR3 gene mutations in the same clinical settings. So it was crucial for us to explore p53-positive part of double-negative UC.

Point 3.

The journal name and doi address of Dadhania's paper (ref. 2) are incorrect. It should be EBioMedicine,  http://dx.doi.org/10.1016/j.ebiom.2016.08.036 (not Oncotarget, ....033)

Reply 3.

We have made appropriate corrections in ref. 2 (lines 583-584). Now it reads ‘Dadhania, V.; Zhang, M.; Zhang, L.; Bondaruk, J.; Majewski, T.; Siefker-Radtke, A. et al. Meta-analysis of the luminal and basal subtypes of bladder cancer and the identification of signature immunohistochemical markers for clinical use. EBioMedicine 2016, 12, 105-117. https://doi.org/10.1016/j.ebiom.2016.08.036 '

Authors

Reviewer 2 Report

In their present contribution, the authors aimed to evaluate how PD-L1 expression in tissue specimens of patients with main molecular subtypes of NMIBC (luminal, basal and double-negative p53-mutant) associates with relapsed-free survival in dependence on the tumor grade and prior treatment of primary bladder cancer. This is relevant since studies on PD-L1 role as a prognostic biomarker and therapeutic target in less aggressive forms of bladder cancer are needed.

Overall, it seems that the manuscript addresses well the proposed goal. The study was well designed, which is reflected on the manuscript. Introduction is appropriate; results are organized, and figures/tables (as well as their descriptions/legends) were carefully prepared, which eases interpretation. Statistical analysis seems appropriate. Discussion complements well the interpretation of the experimental findings presented in the Results. The reference list is appropriate. The main problem with the manuscript is English language and style, that seems confusing and difficult to read and interpret – there’s lot of space for improvement here, mainly in the methods and results sections. The title is also very long and confusing.

Author Response

Response to Reviewer 2 comments

Point 1.

The main problem with the manuscript is English language and style, that seems confusing and difficult to read and interpret – there’s lot of space for improvement here, mainly in the methods and results sections. The title is also very long and confusing.

Dear Reviewer,

On behalf of all the manuscript authors I thank you for positive assessment of our modest work and valuable remarks, which we have used as effective guide to improve our article. We have our manuscript checked by a native American English-speaking colleague, and after that we have made appropriate language and stylistic changes throughout the text with particular focus on ‘Materials and Methods’ and ‘Results’ sections.

Authors
